# Viability Assessment in Liver Transplantation—What Is the Impact of Dynamic Organ Preservation?

**DOI:** 10.3390/biomedicines9020161

**Published:** 2021-02-07

**Authors:** Rebecca Panconesi, Mauricio Flores Carvalho, Matteo Mueller, David Meierhofer, Philipp Dutkowski, Paolo Muiesan, Andrea Schlegel

**Affiliations:** 1Hepatobiliary Unit, Careggi University Hospital, University of Florence, 50134 Florence, Italy; rebeccapanconesi@gmail.com (R.P.); drmauras@gmail.com (M.F.C.); paolo.muiesan@unifi.it (P.M.); 2Department of Visceral Surgery and Transplantation, University Hospital Zurich, Swiss HPB and Transplant Center, 8091 Zurich, Switzerland; matteo-mueller@usz.ch (M.M.); philipp.dukowski@usz.ch (P.D.); 3Max Planck Institute for Molecular Genetics, Mass Spectrometry Facility, 14195 Berlin, Germany; Meierhofer@molgen.mpg.de

**Keywords:** viability testing, machine perfusion, mitochondria, liver transplantation

## Abstract

Based on the continuous increase of donor risk, with a majority of organs classified as marginal, quality assessment and prediction of liver function is of utmost importance. This is also caused by the notoriously lack of effective replacement of a failing liver by a device or intensive care treatment. While various parameters of liver function and injury are well-known from clinical practice, the majority of specific tests require prolonged diagnostic time and are more difficult to assess ex situ. In addition, viability assessment of procured organs needs time, because the development of the full picture of cellular injury and the initiation of repair processes depends on metabolic active tissue and reoxygenation with full blood over several hours or days. Measuring injury during cold storage preservation is therefore unlikely to predict the viability after transplantation. In contrast, dynamic organ preservation strategies offer a great opportunity to assess organs before implantation through analysis of recirculating perfusates, bile and perfused liver tissue. Accordingly, several parameters targeting hepatocyte or cholangiocyte function or metabolism have been recently suggested as potential viability tests before organ transplantation. We summarize here a current status of respective machine perfusion tests, and report their clinical relevance.

## 1. Introduction

Machine perfusion technology is increasingly applied in solid organ transplantation to improve the outcomes through a reduction of ischemia-reperfusion injury (IRI) with less complications and better graft survival [1,2]. The principles of dynamic organ preservation were introduced in 1935 with the Lindbergh apparatus, developed to preserve organs, using a pulsatile recirculation of sterile fluid at normothermic temperatures [3]. Since the first clinical application in the 1960, the technology has now significantly evolved [4]. Two main concepts are currently tested in clinical studies. First, the replacement of static cold storage (CS) by perfusion at normothermic (37 °C) or subnomothermic temperatures (34 °C) with a blood-based perfusate with oxygen [5,6,7]. Second, a concept of organ treatment after CS through perfusion at hypothermic or normothermic/subnomothermic temperatures [8,9,10,11]. A few broad indications for this technology are of interest. Liver perfusion to improve organ quality is also directly linked with the need for a reliable methodology of organ viability testing prior to implantation. To explore the large number of available markers, multiple smaller retrospective case studies with diverse perfusion approaches were performed. As a result, a variety of tests is currently considered helpful to classify livers as viable. Of note, many suggested parameters or thresholds, are frequently based on a small number of clinical liver transplantations, with a lack of systematic analysis of larger or prospective cohorts. Therefore, most authors suggest a combination of “viability” tests, by composite scores based on quantification of different perfusates parameters [12,13,14,15,16,17]. However, as most series have only a limited number of events, the correlation of biomarkers with postsurgical outcome remains difficult [18,19,20,21,22,23]. In this article, we first discuss the clinical relevance of individual parameters, considered for viability testing during machine perfusion before liver transplantation. Second, we highlight certain advantages and disadvantages of suggested markers in context of different perfusion techniques. Finally, we describe the current decision pathways and provide a few suggestions on how to further improve and validate markers currently in clinical use.

## 2. What Do We Expect from a Viability Test?

The term viability carries several definitions in the literature since the earlier days around 1600, when scientists focused on motility to define cellular viability. In biology, viability involves multiple cellular functions dedicated for a single cell or an organism to live, grow, and develop [24]. A more interesting question would be: “What is an appropriate viability test?” or: “How can we determine a proportion of viable cells, tissues, or organs, able to maintain or recover a state of survival?” [24]. Multiple cellular functions are considered to contribute to the status of viability, including cellular morphology, membrane integrity with a functioning barrier, the ability to generate energy to maintain the synthetical function and enzymatic activity, processes of DNA transcription and RNA translation, maintenance of a viable pH gradient and cellular respiration and reproduction [24]. The main task is now to transform such complex measures in a simple and cheap test, which provides enough confidence to accept an organ for transplantation. The more specific a viability test is, the more complex and time consuming is the required methodology. The best example are the concepts of proteomic, metabolomic, and genomic analyses, which enable specified quantification of almost any circulating molecule in a perfusate or tissue [25]. Despite the expected high accuracy, the clinical practicability of such technology, in form of a real-time test with immediate results, appears however relatively low. Based on this, the expectations toward liver viability assessment are very high and include a quick, cheap, and reliable test, which represents the entire organ. Above all, the aim is to identify a validated parameter threshold, to avoid harmful recipient events after transplantation, provided the cut-off is respected.

## 3. Why Is Viability Testing Clinically Relevant?

With an increasing number of candidates, improved medical management and surgical technology, and an aging population with more comorbidities, the quality of donors has decreased. Currently, the cumulative donor–recipient risk accepted by a team, varies among centers and relies on different levels of experience [26,27]. Most countries have applied thresholds for certain risk factors, when to decline, for example a liver from a donation after circulatory death (DCD) donor [28,29]. Selective allocation of organs from extended criteria donors (ECD) is the current policy to avoid severe IRI with immediate complications, including primary non function (PNF) [30,31]. Although, most cut-off values for standard donor and recipient risk factors are based on the outcomes reported from large retrospective cohort studies, their strict application leads to a high number of discarded livers. To overcome this major obstacle in the field and to better understand the metabolic liver environment, scientists have explored the underlying mechanisms of IRI, a complex metabolic phenomenon, which occurs immediately at normothermic reoxygenation of any mammalian tissue [32,33,34]. Importantly, no single parameter has been identified yet, to reliably predict the metabolic status of a specific donor liver and the cellular capability to handle ischemia and subsequent reperfusion injury.

As a first step towards the development of a biomarker, it is therefore important to understand the initial trigger of IRI and the downstream consequences. Looking at different cellular compounds, mitochondria appear at front as instigators of the IRI cascade [35]. During normoxia and aerobic metabolism, mitochondria deliver energy and are crucial for the maintenance of the entire spectrum of cellular functions, considered for a status of viability [24,36]. When tissue is exposed to warm or cold ischemia, first the mitochondrial electron transport and respiration is on hold, based on a complex I–V dysfunction. Subsequently there is a lack of cellular adenosine triphosphate (ATP) [37,38,39]. Second, an accumulation of metabolites linked to the tricarboxylic acid (TCA)-cycle, is seen (Figure 1).

The TCA is blocked through the functional relation with complex II, the lower temperature during ischemia, and the lack of energy to fuel the enzymatic activity [40,41]. At subsequent rewarming, the four mitochondrial complex proteins restart the electron transfer with an undirected electron transport and the subsequent production and release of reactive oxygen species (ROS) from complex-I (Figure 1).

ROS release appears immediately within the first few minutes after reoxygenation [32,40,42,43,44,45]. Importantly, the level of complex-I injury strongly depends on the temperature applied during reoxygenation. Significantly higher levels of mitochondrial injury and dysfunction were found during normothermic reoxygenation, when compared to hypothermic reoxygenation [38]. The substantial recovery from energy loss during ischemia was quantified by many historical studies through the measurement of high tissue ATP concentration’s after a few hours of liver re-oxygenation under cold conditions [37,43,46,47,48]. The full picture of IRI develops with further downstream inflammation, quantified by Damp- and cytokine release from all cells in perfusates and recipient blood [43,49]. Livers, which undergo hypothermic reoxygenation on a device (hypothermic oxygenated perfusion; HOPE/D-HOPE, HMP-O2), experience reprogramming of the mitochondrial respiratory chain with ATP reloading (Figure 1). As a consequence, such organs are protected from IRI features, including ROS-, Damps-, and cytokine release and tissue inflammation at subsequent NMP or transplantation [50,51]. Normothermic machine perfusion (NMP) applied after static cold storage simply exposes the entire picture of IRI and therefore enables the measurement of various molecules released into perfusates. This is considered beneficial by many to develop a test to assess liver viability during NMP [6]. Two strategies appear. First, identify a large number of unspecific parameters and thresholds, requiring artificial intelligence to calculate sort of score points to decide, if a single organ could be used for transplantation [52]. Or second, an easy, cheap, and more precise approach using a few single tests of biomolecules, released from mitochondrial complex I, the key site, where the IRI cascade initiates [40,53,54].

## 4. What Are Available Modalities to Test Viability?

Liver viability assessment starts at the time of donor offer, where the surgeon draws the initial picture during the first conversation with the involved donor coordinator prior to macroscopic examination. With the delivery of the donor past medical history, results of donor blood tests, including liver transaminases, parameters of coagulation, cholestasis, hypoxia and electrolytes, the viability testing begins (Figure 2). In some countries further donor imaging studies are allowed, to perform abdominal ultrasound when for example liver steatosis is expected (Table 1). During procurement surgery organ shape, size, color, flush, and vessel quality are routinely explored. With the available modern technology, results are immediately communicated to the recipient center and further tests are subsequently initiated. For example, liver biopsies are performed at the donor center or sent with a combined transport to the recipient for analysis, or obtained directly by the recipient team when the organ is unpacked from the ice box [55,56]. Unless a liver undergoes continuous machine perfusion on a perfusion device, there are limited opportunities to assess the viability further during cold ischemia and transport (Table 1 and Figure 2).

Machine perfusion technology currently evolves significantly, and two main concepts are presently introduced in clinical practice. First, perfusion instead or after limited cold ischemia during transport of the liver and second, an endischemic approach of various machine perfusion concept after initial cold ischemia and transport to the recipient center. Both, normothermic machine perfusion (NMP) or hypothermic machine perfusion (HMP) are beneficial for organ assessment because the recirculation of an oxygenated perfusate uncovers various molecules, released from all liver cells into the perfusate at all temperatures.

Routine blood gas or biochemical analysis were transferred into the field of organ preservation and have been performed either from the cold storage solution or machine perfusates [10,57,58,59]. The majority of markers is however unspecific, being copied from clinical practice of patients with liver diseases, including for example pH, lactate, or liver transaminases. Further potentially useful tests include perfusate analysis with spectroscopy to quantify specific markers, released from the subcellular compounds (Table 1, Figure 2 and Figure 3).

Although this technology is also not new, it was recently implemented to quantify molecules used for viability assessment during HOPE treatment [23,38]. Other than perfusate or flush solution, the fluid released through the biliary tree during NMP is used to indirectly measure hepatocyte and cholangiocyte injury or function. Bile parameters include pH, bicarbonate and glucose concentration, and lactate dehydrogenase (LDH) [19,60]. Perfusates, bile and tissues can be obtained throughout the entire time an organ is exposed to dynamic perfusion modalities. A large body of tests, including the LIMAX test, the quantification of miRNA, mitochondrial DNA or Damps and cytokines, metabolomic/proteomic/genomic analyses, and ATP quantification, could be applied [25,61,62,63]. Clinical use of such modalities is limited by the prolonged time required to receive a result, or the need for tissue biopsies, where a systematic analysis to identify the best marker combination is still lacking today.

## 5. What Is More Important: Liver Injury or Function?

Livers with more than 108 cells per gram weight are the metabolic power house in our bodies and accomplish more than 500 functions [64]. Perfusate analysis with metabolomic technologies reveal several thousand molecules, released from one single organ. Such molecules can be classified according to their origin in different cellular compounds including cytosol, mitochondria, endoplasmic reticulum (ER), or nucleus (Figure 3). In addition, various protein classes, including damps, cytokines, chemokines, kinases, and heat shock proteins can be used to group those molecules, known from experimental studies as markers to describe the level of IRI [43,65,66,67]. The challenge is now, how to best identify those parameters, which represent the metabolic liver function under healthy conditions, and are also relevant for viability testing during machine perfusion. The best example are liver transaminases, routinely measured to explore the level of liver injury before and after liver resection or transplantation, both aspartate-aminotransferase (AST) and alanine-aminotransferase (ALT) appear very unspecific with a limited predictive value [68]. Similarly, perfusate or plasma damps and cytokines are frequently measured in experimental studies to describe the tissue response to injury or to a specific treatment. The required technology with ELISA or PCR appears time consuming with several hours to day of analysis and are less practical in the setting of machine perfusion.

Hibernators induce changes in their cellular respiration through mitochondria, which further highlights their relevance. Of note, hibernating animals are naturally protected from IRI during periods of arousal, when returning to normal cellular functions and temperature [69,70]. Based on this and the known key instigator of IRI, situated in mitochondria, such compounds appear at front to describe the function or viability of an organ, including the liver [36,45,71]. This is further supported by Sumimoto et al. who nominated mitochondrial ATP production as a reliable marker for viability in 1988 [72]. Energy is of utmost importance for any cellular process and highlights the need for proper mitochondrial function. Today, the quantification of cellular ATP requires several hours and thereby does not represent a useful marker for real-time viability testing, while an organ is on a perfusion device. Measurement of surrogate markers for ATP recovery during machine perfusion, which represent the metabolic situation of the complex I and II in the respiratory chain, is needed.

## 6. How Do We Test Viability during Normothermic Machine Liver Perfusion?

Normothermic machine perfusion (NMP) is nominated by many as the best perfusion approach to assess organ viability [59,73]. Various authors claim this technology provides “near-physiological” conditions, although the question, what circumstances from organ donation, transport and transplantation are really physiological? appears [9,10,74].

Two main groups of parameters, selected as “viability criteria,” are in clinical use during NMP to differentiate between liver cells (hepatocytes) and the biliary tree. Additional parameters of perfusion quality, metabolic, and excretory organ function are used to explore how viable a specific organ appears. In 2020, Raigani et al. have demonstrated the median costs of $28,099 USD needed to identify a potentially transplantable liver with NMP in the US [17].

### 6.1. Perfusion Quality and Hemodynamic Parameter

Parameters of perfusion quality are routinely considered as surrogate marker for liver function. Vascular resistance contributes to perfusion flows, which are also dependent on perfusion pressure. An increased vascular resistance has been linked with a generally impaired liver function [75]. Livers with significant levels of macrosteatosis may experience narrowing of the sinusoids with subsequent reduced perfusion flows [76]. One consequence of sinusoidal obstruction due to fat droplets or to stuck blood cells during reperfusion is a reduced flow, which leads to secondary hypoxia. These features induce an additional aggravation of IRI with ROS, Damps, and cytokine release and ongoing inflammation with organ dysfunction. Perfusion quality is therefore an important parameter. High resistance has been a sign of organ injury and later dysfunction. Stable perfusion flow rates during NMP were therefore suggested by many [22,59,60]. A few authors have defined more specific viability criteria with a portal vein and hepatic artery flow above >500 mL/min and >150 mL/min, respectively (Table 2) [9,57,74]. Some case series described a reduction in flow rates toward the end of prolonged NMP, a sign of sinusoidal exhaustion and advanced injury [12]. However, when an impaired flow and high resistance is present, an advanced injury is usually seen on histological examination. Parameters of perfusion quality are therefore rather late signs of a failing organ, which might explain, why most studies failed to discriminate between viable and non-viable livers based on the perfusion flows only [20,60,77].

### 6.2. How Helpful Is the Quantification of Liver Transaminases?

The level of transaminases measured in perfusate is another unspecific parameter with many confounders. Prolonged perfusion first, induces hemolysis and leads to additional AST release from erythrocytes. In contrast, a very advanced-injured liver does not release more transaminases into perfusate, due to the “empty hepatocytes,” interpreted as “false low.” Cut-off values are largely missing and those suggested by various groups appear arbitrary. Clinically applied AST and ALT thresholds range from 6000 to 9000 U/L (Table 2) [21,60,78]. Ceresa et al. have discarded one graft in their trial based on an ALT of 9268 U/L in the first hours of NMP [21]. The group from Innsbruck has declined 4 grafts due to an “inadequate high perfusate AST” [78]. Livers with very high transaminase release are quickly classified as “non-viable,” while low values are unspecific, and both scenarios are challenging. Although transaminases could be normed to liver weight and perfusate fluid to improve the predictive value, their correlation with outcome parameters was always weak in clinical studies [68].

### 6.3. What Is the Predictive Value of Lactate?

Although lactate is the most prominent parameter, used to assess the viability during NMP, the predictive value is limited. The key question here is to understand, why we fail to distinguish between viable and non-viable livers based on perfusate lactate measurements? During warm or cold ischemia, mitochondria experience hypoxia and respiratory failure with subsequent switch to anaerobic glycolysis. Lactate is produced from pyruvate by lactate dehydrogenase under anaerobic conditions (hypoxia).

Importantly, lactate metabolism or clearance to glucose or glycogen requires oxygen and ATP [79]. During normothermic reperfusion in the recipient or on a perfusion device, lactate measurements show a curve with three phases, an initial high peak with a fairly rapid clearance, followed by a maintenance phase with stable and low lactate values between 2 and 4 mmol/L. Lactate clearance mainly occurs in periportal hepatocytes, representing zone one of the liver lobule, which is the last zone exposed to oxygen deprivation. In contrast, hepatocytes in zone two and three, both with more distance from the portal triad, contribute less to lactate clearance. Such variations in liver metabolism further support recent findings, that a persistently elevated lactate is a sign of advanced, panlobular injury and liver dysfunction in a late stage or an ongoing lactate production from hypo-perfused parenchyma [58].

Although ex situ perfusion circuits include only the liver, lactate is also released from erythrocytes, particularly when the perfusion duration is prolonged (more than 24 h) [64]. An additional challenge is the multiple suggested thresholds for lactate in the literature. Based on experimental studies that identified a combination of perfusate lactate, bile production, and stable perfusion flows as “reliable” predictors of graft viability, the group from Birmingham initially suggested lactate as the key parameter. In 2016, Mergental et al. published the outcome of 5 human livers, transplanted after NMP. Liver grafts were accepted when lactate clearance to ≤2.5 mmol/L was achieved within the first 3 h of NMP or when the organ met a combination of other parameters, including bile production, perfusion quality, and perfusate pH of >7.3 [80].

Next, the same group presented a cohort study of 12 livers. Grafts were divided into a lactate clearing (LC)- and a non-LC group, where lactate levels showed constant fluctuations until the end of 6 h NMP [14]. Authors performed explicit liver and perfusate assessment without transplantation and demonstrated a link between lactate clearance and bile production, ATP recovery and more healthy histological features.

Based on a clinical experience with their viability criteria in 5 human liver transplantations, the Birmingham group started a new clinical trial, using lactate clearance in combination with other parameters to confidentially decide if livers qualify for transplantation. Within the VITTAL trial, 31 human livers underwent NMP and 22 grafts were transplanted, based on the following criteria: the achievement of lactate clearance to <2.5 mmol/L, combined with a perfusate pH of >7.3, the evidence of bile production, a homogenous liver perfusion, glucose metabolism and portal vein flow of ≥500 mL/min, and hepatic artery flow rates of ≥150 mL/min (≥2 of the criteria are required to achieve “viability”) [9]. Of note, in this trial the assessment period was prolonged to 4 h to increase the number of utilized livers. Although combined with other parameters, lactate clearance was not predictive enough to protect recipients from the development of 4 ischemic type biliary lesions (ITBL) [9].

In contrast, Reiling et al. applied a threshold of ≤2.0 mmol/L for perfusate lactate, required within the first 2 h of NMP [59]. Finally, the Groningen group suggested a cut-off ≤1.7 mmol/L lactate during NMP in their cohort studies (Table 2) [81]. Most suggested thresholds appear arbitrary, because of the lack of events, which confirm that transplantation of livers with perfusate lactate beyond such thresholds leads to severe complications, such as PNF (Table 2). Unfortunately, this paradigm is valid for most perfusion parameters, suggested to provide confidence to select livers for implantation.

Another challenge is the timing, when to best measure lactate during perfusion. The various timepoint when to decide to accept a liver or not with 2, 2.5, 3, or 4 h of NMP appear randomly selected due to a lack of sufficient data. Interestingly, in their initial paper published in 2016, the group from Birmingham suggested to quantify lactate after 3 h of NMP, then moving to 2.5 h and in their recent VITTAL trial the latest recommendation was 4 h of NMP [9,14,80].

Following initial lactate clearance, Ceresea et al. have used a second threshold of increasing lactate to ≥4 mmol/L during later perfusion as “warning sign” and discarded 2 livers [21]. Two human livers were also declined in the VITTAL trial because of an increasing lactate after meeting the initial 2 h criteria [9]. This is paralleled by the frequently remaining high blood lactate, despite continuous hemofiltration in patients with severe acute liver failure. Based on this, lactate is considered as rather late and irreversible sign of significant dysfunction and cell death [82].

Next, individual testing of dialysis components in sophisticated perfusion devices used for prolonged NMP could not achieve lower lactate levels of less than 2.5 mmol/L. These findings underline the fact that even metabolically severely impaired livers have a remaining ability to clear a certain amount of lactate. A study by Watson et al. investigated perfusion characteristic during NMP of 22 transplanted livers. Lactate declined to ≤2.5 mmol/L in all but 5 human livers within two hours. Interestingly, the only liver that suffered a PNF after transplantation, reached this threshold within 90 min of NMP [58,60].

Despite the frequent use of lactate as the “key parameter” to select viable livers, it is a rather unreliable and unspecific marker to test the quality of a specific organ. Of note, Nasralla et al. have observed a PNF in the perfusion arm of their randomized controlled trial, despite sufficient lactate clearance during NMP according to the above mentioned criteria [10].

In this context, the molecule lactate may be more beneficial in predicting organ function when measured more frequently using the area under the curve (AUC), instead of single values. Additionally, each viability parameter, including lactate should be quantified in a more systematic, prospective approach and in a higher number of cases. Third, measured values might be of more relevance when normed to the liver weight and the amount of circulating perfusate. Although lactate represents the anaerobic glycolysis, other markers including cumulative purine metabolites and succinate may provide a more specific picture of oxidative mitochondrial function.

### 6.4. Perfusate Acid Balance and pH

Along with advanced donor liver injury or ischemia times, and subsequent tissue hypoxia and anaerobic metabolism, a low perfusate pH is usually seen during NMP. The majority of case series recommend to maintain a physiological pH range between 7.3 and 7.45 [19,20,74,80,81]. Many other factors have however impact as confounders, including the perfusate composition and additives. For example, bicarbonate, also used as surrogate marker for viability, is routinely added to perfusate during NMP. The bicarbonate bolus is routinely administered at start of NMP according to perfusate pH, and ranges between 10 and 40 mmol/L [21]. Watson et al. described in their series, that one liver requiring the administration of 30 mmol bicarbonate during NMP, more than any other graft, has developed a PNF after implantation [60]. Authors have therefore included a maximal bicarbonate bolus of ≤30 mmol/L to maintain a perfusate pH of ≥7.2 in their criteria (Table 2) [60]. Based on multiple confounders, including the partial pressure of carbon dioxide and the bicarbonate consuming urea production, perfusate pH and acid balance are considered only in combination with other cellular parameters to determine liver viability. False high or low perfusate pH values could be the consequence with transplantation of riskier organs or the loss of viable livers deemed of too high risk.

### 6.5. Glucose Metabolism

To mobilize energy from long-term storages, cells switch to anaerobe metabolism when ischemia occurs. The lack of ATP leads to more glycogenolysis, which can be measured through elevated glucose levels in perfusates during NMP. Reiling et al. have included the perfusate glucose concentration at 4 h of NMP in their criteria (Table 2) [59].

The stimulation of gluconeogenesis, while blocking glycogenolysis was recently suggested as supportive test [12,58,60]. However, a few confounders need to be discussed. When organs experience severe ischemic injury, glucose can be measured equally low as in metabolically active organs. Based on this, glucose challenge tests have been suggested for viability testing. The addition of glucose to the perfusate is performed to trigger gluconeogenesis, which is only active in cells, capable to perform aerobe metabolism with gluconeogenesis. With a significant proportion of healthy hepatocytes in the liver, the perfusate glucose level decreases [9,12].

Viable livers also metabolize glucose in response to insulin administration, another surrogate for mitochondrial function. The group from Zurich is therefore in favor of such responsive tests to external triggers (such as inulin or vasoactive drugs) instead of simple measurement of single factors. Such challenging tests provide better information particularly when combined with other analyses compared to lactate alone [12,64].

### 6.6. How to Assess the “Biliary Tree” during Machine Perfusion?

Both, healthy hepatocytes and cholangiocytes, contribute to the production and composition of bile in the liver [83]. The large number of hepatocytes actively release bile acids, an ATP-dependent process, which depends on mitochondrial function (Figure 3) [16]. Cholangiocytes release bicarbonate and absorb glucose from bile [84,85]. In addition to donor liver quality and the level of IRI, the volume of secreted bile during machine perfusion depends on a proper placement of the bile duct cannula, which can kink and block the bile flow. False low bile flow counts are the consequence.

The group from Groningen suggested a minimal target bile flow of 10 mL in the first 2.5 h [81]. However, graft loss has been described despite proper bile production [58,60]. Ceresa et al. found 4 livers with no appropriate bile flow within 4 h of NMP, all were transplanted and showed immediate function and no relevant graft quality-related complications (Table 2) [21]. The four livers, transplanted by Zhang et al. after assessment during NMP showed immediate full function and would have been declined according to the Groningen criteria, because none achieved 10mL bile production within the first 2.5 h of NMP [74]. Such results are further supported by the VITTAL study, where all transplanted livers achieved hepatocyte viability criteria including bile production. However, 45% of recipients developed some form of irregularities in the bile ducts, seen in the magnetic resonance cholangiography (MRCP). Of note, 18% (n = 4/22) required retransplantation for ITBL [9]. The authors of the VITTAL study therefore recommend to avoid transplantation of human DCD livers with prolonged donor warm ischemia time after preservation with endischemic NMP [9]. Next, the group from Cambridge has assessed the utilization of 12 discarded, human livers with endischemic NMP. Despite the achievement of hepatocyte viability, 25% of recipient developed an ITBL [58].

The majority of studies, which aimed to assess liver viability, focused on hepatocytes. In context of the ITBL incidence, ranging between 2.6 and 33.3% in the last 10 years of DCD liver transplantation, the interest in cholangiocyte function is increasing [86,87,88]. Although bile flow is essential to enable assessment of the bile fluid, the predictive value as a single marker to explore the quality of the bile ducts is limited [19]. Cholangiocytes line the main branches of the biliary tree and contribute significantly to the composition of bile, based on their bicarbonate secretion and glucose absorption (Figure 3) [84,85]. In their first series of 12 livers, the group from Cambridge measured the biliary pH during NMP, and although it was not considered for decision-making, authors were the first to suggest bile pH as relevant parameter to test viability and the group associated low values with the three ITBLs in their cohort [58].

Three different cut-offs for bile pH during NMP were suggested in the last 3 years. In their second and main systematic liver viability analysis, the group of Chris Watson has included 47 human livers. Three out of 16 transplanted livers, did not achieve a biliary pH of >7.4 and developed an ITBL [60]. Based on 23 human livers, which underwent NMP, Matton et al. have suggested a different bile pH threshold of >7.48. This cut-off was applied in 6 additional perfusions, where 4 livers met criteria and underwent successful transplantation [19]. Four month later, the same group from Groningen however suggested to use a different threshold of bile pH with >7.45 at 2.5 h in a cohort of 11 DCD livers, transplanted after NMP (n = 11/16) (Table 2). Of note, authors applied here a different concept and combined a graft treatment with D-HOPE, followed by controlled oxygenated rewarming (COR) plus viability assessment during NMP. In addition to bile pH, biliary glucose concentrations of <16 mmol/L and a bicarbonate level of >18 mmol/L should be achieved within 2.5 h of NMP by livers selected for transplantation [19]. Although all 11 implanted livers met both, hepatocyte and bile duct viability criteria, one ITBL was seen [81]. Based on this, authors suggested now to follow ratios and deltas between perfusate and bile, instead of absolute values for bicarbonate and glucose [81]. The calculation of biliary over perfusate glucose concentration ratio (<0.67) at 2 h of NMP, or subtraction of biliary glucose from perfusate glucose at 3 h, were considered [19]. Biliary LDH concentrations of <than 3689 U/L within 2.5 h of NMP were further nominated as viability tests for the biliary tree [19]. Livers that met such criteria, were also found with a low bile duct injury (BDI) score on histological examination [19]. Although these are promising steps toward the identification of more reliable markers, the number of overall transplanted grafts remains low (Table 2).

### 6.7. Clinical Decision Making Based on Viability Parameter

Almost each group, or center and surgeon apply not only different perfusion techniques, but also various criteria or parameter thresholds. For example, in Birmingham, livers are routinely assessed macroscopically in the recipient center and undergo NMP, in context of a clinical study and provided the team in favor of this perfusion technology agrees. Specific criteria are then used to accept a liver for transplantation within the first 4 h of NMP [9]. The lactate threshold used is ≤2.5 mmol/L, while the team from Australia considers a perfusate lactate level of 2 mmol/L as cut-off [59] and the group from Cambridge accepts livers when a peak lactate fall of ≥4.4 mmol/L/kg/h is seen [60]. Similar features apply for all other parameters listed in Table 2. An overview of clinical decision pathway comparing normothermic and hypothermic perfusion approaches with subsequent viability parameters is provided below. Of note, none of the studies has considered different tests, parameters, or thresholds according to the type of graft or the donor age. A generally more severe IRI with subsequent graft dysfunction is reflected by a number of different parameters, which are currently not specific enough to discriminate between metabolically impaired old or young donor grafts or different levels of steatosis. The vast majority of viability test is based on the analysis of DCD liver grafts and subsequently generalized on other graft types.

Such examples of relatively small cohort studies with different perfusion conditions applied during NMP, various molecules tested in perfusate and bile, multiple cut-offs and timepoints suggested for the measurement of individual parameters, impressively demonstrate the enormous variations and confounders in this field. In addition, the work load ahead of us is obvious to identify the most accurate parameter threshold to support a confident decision during NMP. In 2018, Peter Friend already summarized the current state of viability assessment during NMP as follows: “Data from much larger numbers of transplants done with normothermic perfusion (typically from a registry) would be required to determine specific markers of viability.” [10].

## 7. Viability Assessment during Hypothermic Oxygenated Perfusion

### 7.1. Can We Use the Same Parameters as in Normothermic Liver Perfusion?

Hypothermic oxygenate perfusion (HOPE) is routinely applied after cold storage through the portal vein only (or dually through PV and HA; D-HOPE; HMP-O_2_) with an artificial and high dissolved oxygen concentrations at 8–10 °C [89]. Results from six ongoing randomized controlled trials on cold liver perfusion are currently awaited. Although hypothermic technologies are frequently considered less helpful to assess liver viability, the same molecules found in NMP perfusates can also be identified during cold or subnormothermic liver perfusion (Figure 1, Figure 3 and Figure 4) [18,23,38,50,65,90].

Similarly to NMP, perfusion flow, pressures, and resistance are routinely measured, but rarely considered as single criterion for viability testing [1,8]. A few retrospective studies have explored the value of the same markers used for viability testing during NMP, in hypothermic perfusates. The predictive value of perfusate lactate, ALT and LDH was assessed systematically in a high-risk DCD liver cohort from Switzerland. Lactate and transaminases were equally ineffective in predicting posttransplant outcomes, similarly to different risk scores, including donor risk index (DRI) and liver graft assessment following transplantation risk (L-GrAFT) score [23]. Perfusate transaminases correlated with posttransplant recipient transaminases, with however no predictive value. Such findings parallel earlier studies, where similar correlation between perfusate and recipient plasma transaminases were demonstrated with however limited further impact on complications [90,91].

Recently, Patrono et al. published the results of a single-center retrospective study on 50 patients, who received a DBD liver transplantation after D-HOPE [18]. Their data showed a negative correlation between lactate and pH explained by the fact that lactate accumulates during CS as a product of anaerobic glycolysis and when it is subsequently released in the perfusate it determines a reduction of pH. Of note after normalization to liver weight, lactate lost the correlation with markers of hepatocyte and cellular injury (Table 2). In addition, perfusate glucose concentrations failed to predict graft survival during D-HOPE, and lactate was the only perfusate parameter not predictive for EAD [18,23]. During cold perfusion there is also fluid secretion through the biliary tree, corresponding to perfusate mixed with molecules released from hepatocytes into bile, which flow through and between hepatocytes into the sinusoids and ductuli’s, following the regular pathway of bile. Recently, the Zurich group has demonstrated a completely fluoresceine-stained liver including the tip of the common bile duct, in the first 5 min after administration of this dye into the perfusate. Importantly the HOPE was performed through the portal vein only, paralleled by the known venous collateral mesh, fed by the portal vein, and surrounding the common duct [92,93]. However, in the cold there is a lack of active secretion of bile acids and other molecules, leading to clear fluid in the biliary tree. This fluid has not been systematically collected and assessed yet.

### 7.2. How Can We Assess Mitochondrial Function and Injury?

The protective effect of hypothermic reoxygenation on mitochondrial function with energy recovery is known from many historical studies [46,94,95,96]. Important mechanistical differences comparing normothermic and hypothermic reoxygenation were described. Organs exposed to warm and cold ischemia unveil their mitochondrial injury at reperfusion or “reoxygenation” through ROS production and release at complex I, with downstream inflammation of the entire surrounding tissue. Metabolomic perfusate analysis identified a specific protein Flavinmononucleotid (FMNH_2_), released from NDUFS1–a “key pocket” in mitochondrial complex-I, the same localization of ROS production [34,40,53,54,97,98]. During physiologic conditions, FMN is tightly bound to complex I. When tissues experience ischemia with subsequent reoxygenation, a certain amount of FMN is released into perfusate or blood, because of the interrupted mitochondrial respiration and the reduction of the ubiquinone pool [99]. The perfusate FMN concentration depends on the liver quality and the temperature applied during reoxygenation. Similarly to the higher level of IRI and ROS, seen during normothermic reperfusion when compared to hypothermic reperfusion, FMN release from complex I is significantly lower under cold conditions (Figure 1) [38]. Galkin et al. have recently demonstrated the importance of the FMN pocket for the ROS production when tissue undergoes normothermic reperfusion [98]. Structural alterations of the pocket may also impact the affinity of FMN and possibly ROS molecules within the mitochondrial complex I.

While the great importance of mitochondrial respiration and energy production is known since more than 30 years, the impact of perfusate FMN as surrogate marker for cellular energy production (ATP) and viability appears new [72]. However, the autofluorescence abilities of FMN have been described in 1969 [100]. High FMN perfusate concentrations correlate with high levels of electron donors and the ATP reduction to precursors, measured by purine metabolites in tissue and perfusates [38]. In contrast to conventional perfusate parameters, such as lactate and transaminases, which failed to predict outcomes after liver transplantation, perfusate FMN concentration correlates with liver function, EAD, hospital stay, cumulative complications, and most importantly 3-month graft loss after liver transplantation with a very high accuracy [23,38]. Muller et al. have established a clinically useful threshold of perfusate FMN. If the concentration climbs above 8800 A.U. at 30 min of HOPE or a sharp incline is seen, authors recommend not to use the liver for implantation. Retrospective analysis has revealed this threshold and 67% of livers with perfusate FMN levels beyond the threshold were lost (Table 2) [23,38]. Importantly, Guarrera et al. have confirmed the correlation of FMN with posttransplant liver function in perfusates of their ongoing RCTs [101].

In clinical practice, any donor liver with extended risk (ECD, DCD) offered is initially accepted and undergoes HOPE-treatment after standard procurement and transport. The perfusate FMN concentration at 30 min of HOPE treatment is considered the most important variable. If perfusate FMN values are found below 8800 A.U., the liver is accepted for the initially allocated recipient, independent of the recipient’s disease severity or MELD score. If there is a sharp FMN incline, a repeat FMN measurement at 45 or 60 min of HOPE perfusion is recommended. If the perfusate concentration climbs further beyond the suggested cut-off, the organ is discarded. If there are high but stable FMN perfusate values below the threshold of 8800 A.U., the organ is reallocated to a less sick recipient. Of note, each center follows their specific policy of donor and organ acceptance with various indications for machine perfusion, particularly when applied as compassionate use or outside any clinical trial. Currently used parameters are not assessed for their specificity to discriminate between steatotic livers or DCD grafts for example. However, the FMN threshold described above is valid for DBD grafts and ECD livers of all types including steatotic grafts (Figure 5) [38].

Based on the tight connection between FMN release and IRI, both starting from the same pocket in complex I, FMN release could also be a useful marker for viability during other perfusion approaches. Wang et al. have now demonstrated the impact of FMN as a biomarker during normothermic kidney perfusion and normothermic regional perfusion (NRP) in DCD donors [102]. Perfusate FMN predicted posttransplant renal function and the FMN value at 30 min of NRP may support clinicians to accept DCD donors or not [102].

During ischemia the electron transport across mitochondrial complexes is impaired and next to succinate, another molecule, NADH accumulates. The metabolic reaction of NADH to donate protons for the H+ gradient and electrons, both for the ATP production is linked within the same area in complex I. An improved complex I function through HOPE treatment, leads therefore to increased NADH metabolization. Similarly, to FMN, the molecule NADH has autofluorescence abilities and is used for viability assessment. Already in 2013, the link between perfusate NADH concentration and mitochondrial metabolism was demonstrated [103]. Confirmed through mass spectrometry analysis, the threshold of 10,000 A.U. perfusate NADH was established at 30 min of HOPE and is routinely used in combination with FMN cut-offs to provide a more accurate assessment of mitochondrial complex I function and injury (Table 2, Figure 5) [38]. As with all used markers for viability, NADH was described earlier. Van Golen et al. have shown the importance of the autofluorescence abilities to determine viability as a surrogate for mitochondrial function and cellular energy [32]. Several multicentric clinical and experimental studies are currently ongoing with different perfusion techniques to further validate both complex I markers. The FMN pocket in complex I (NDUFS-1) was recently immuno-stained. Interestingly, a fully functional mitochondria complex I appears colored, while in liver cells with advanced complex I injury the NDUFS-1 unit is released into the cytosol [38].

## 8. Summary and Future Perspective

Multiple parameters to assess viability during machine perfusion have been suggested within the last 5 years. Today there is no widely accepted marker, which is reliable to provide enough confidence to accept a liver [52]. Two potential strategies are available. The identification of multiple parameters with complicated assessment, requiring computerized artificial intelligence to decide if to accept an organ. Or a simple key parameter, which is linked to the central mechanism of organ injury initiated in the mitochondria. A few steps are crucial to improve the current situation: first, the general agreement to store perfusate and tissue from any machine perfused liver in a local or central biobank. The collection of larger sample sizes allows to merge samples and to perform a systematic screening for biomarkers. Next, results from expected randomized controlled trials including livers with higher risk will hopefully reduce the differences among perfusion protocols, which include perfusate composition, perfusion route, duration, device used, supplements etc. This is very important to achieve more specific and generalized data. Third, technical advances will help to obtain results from miRNA detection, ATP quantification, and metabolic profiling within shorter time. Above all, collaborative approaches are needed to merge the cases, perfusion samples, and ideas to identify one marker, which is quick, cheap, and ideally assessable in real-time during perfusion with all devices and at all temperatures.

## Figures and Tables

**Figure 1 biomedicines-09-00161-f001:**
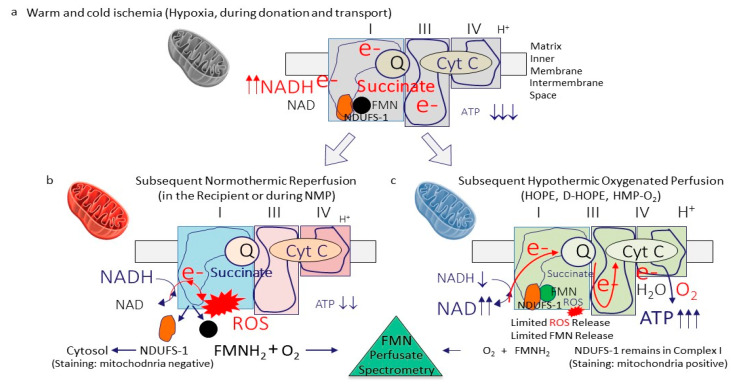
Role of mitochondrial complex I as an instigator of ischemia-reperfusion injury. During any sort of warm or cold ischemia, the mitochondrial respiration is on hold and leads to a lack of cellular ATP with metabolite accumulation, including succinate. At the same time, the complex-I electron donor NADH accumulates (**a**). When oxygen is reintroduced, mitochondrial respiration is reinitiated, with a however incongruent speed of the different complexes 1–4, leading to the development of reactive oxygen species (ROS), the initiator molecule of further downstream tissue inflammation. Flavin-mononucleotide (FMNH_2_) are released from the same area in complex I, next to the ROS molecules. Together with damps and cytokines, such molecules react with oxygen and are released into the circulation, where a spectrophotometric quantification of FMN is possible. This complex 1 injury is significantly more pronounced during normothermic reoxygenation (reperfusion) (**b**), when compared to reoxygenation under hypothermic conditions (**c**). The higher the complex 1 injury, the more ROS and FMN molecules are released. Interestingly, NDUFS-1, the complex I FMN pocket, disconnects and is released into the cytosol, where staining procedures are used to quantify the mitochondrial complex 1 injury. ATP: adenosine triphosphate; NAD/NADH: nicotine adenine dinucleotide (oxidized/reduced); ROS: Reactive oxygen species; FMN: flavin mononucleotide; NDUFS-1: NADH: ubiquinone oxidoreductase core subunit S1 (mitochondrial complex I-subunit); O_2_: oxygen; Cyt C: cytochrome C; e-: electron; HOPE: hypothermic oxygenate perfusion; D-HOPE: dual hypothermic oxygenate perfusion; HMP: hypothermic machine perfusion; NMP: normothermic machine perfusion.

**Figure 2 biomedicines-09-00161-f002:**
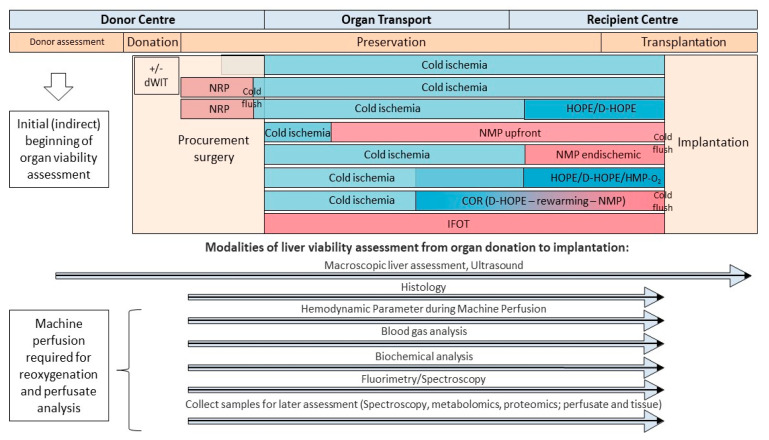
Timeline and modalities to assess viability from organ donation, during preservation, and transplantation. Various tests and modalities contribute to the overall picture of viability assessment during organ preservation. Dynamic preservation methods provide various advantages to sample perfusate, bile, and tissue for parameter quantification. dWIT: donor warm ischemia time; NRP: normothermic regional perfusion; COR: controlled oxygenated rewarming; HOPE: hypothermic oxygenate perfusion; D-HOPE: dual hypothermic oxygenate perfusion; HMP: hypothermic machine perfusion; IFOT: ischemia free organ transplantation; NMP: normothermic machine perfusion.

**Figure 3 biomedicines-09-00161-f003:**
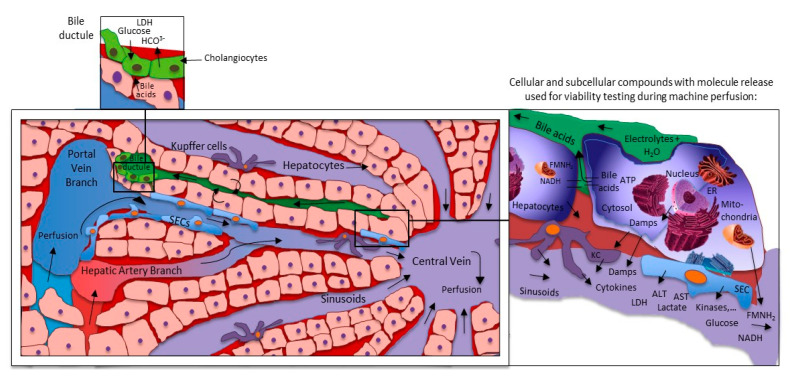
Liver machine perfusion and marker release on cellular and sinusoidal level. Every cellular compound release molecules of injury during reperfusion. Hepatocytes appear at front, also due to their number in the liver. The majority of molecules are simply washed out into the circulation or perfusates. Only very few represent the metabolic situation of the organ and the task is their identification. ER: endoplasmic reticulum; KC: Kupffer cells; SECs: sinusoidal endothelial cells; LDH: lactate dehydrogenase; HCO^3−^: monohydrogencarbonate; ALT: alanine aminotransferase; AST: aspartate-aminotransferase; ATP: adenosine triphosphate; Damps: danger associated molecular patterns; NAD/NADH: nicotine adenine dinucleotide (oxidized/reduced); ROS: reactive oxygen species; FMN: flavin mononucleotide; NDUFS-1: NADH: ubiquinone oxidoreductase core subunit S1 (mitochondrial complex I-subunit).

**Figure 4 biomedicines-09-00161-f004:**
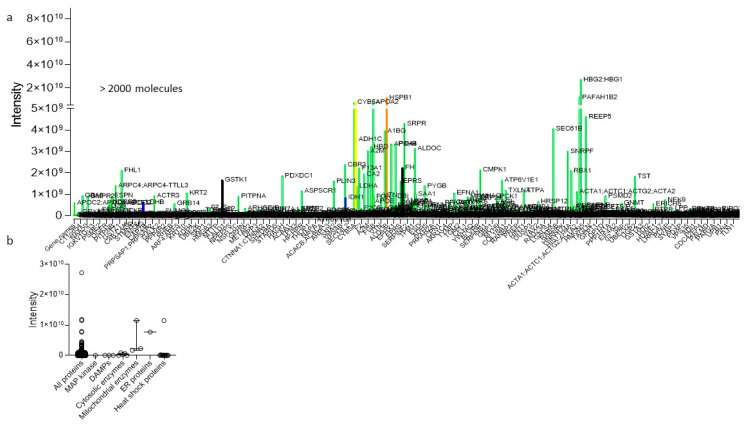
Mass-spectrometry-perfusate analysis, obtained during hypothermic oxygenated perfusion (HOPE). Numerous molecules are identified through mass spectrometry from perfusates obtained during HOPE-treatment of human livers (**a**). Such molecules were allocated to specific molecule classes released from any cell type (**b**). HOPE: hypothermic oxygenated perfusion.

**Figure 5 biomedicines-09-00161-f005:**
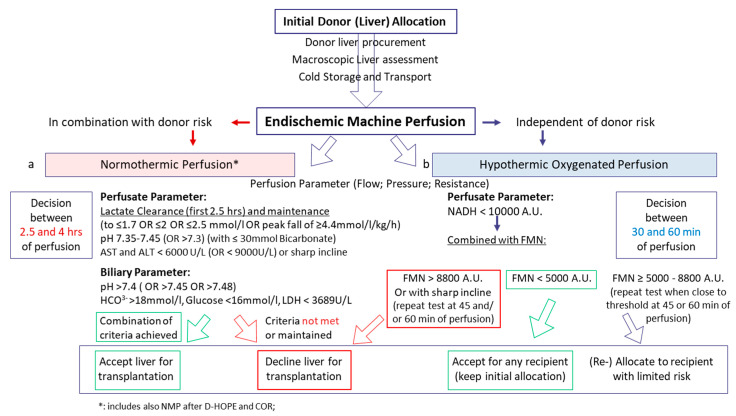
Pathway on clinical decision-making based on viability assessment during endischemic machine liver perfusion. Dynamic liver perfusion serves as a tool to test organ quality and to predict function and outcomes after transplantation. Various parameters were suggested for routine clinical use. During normothermic perfusion, parameters of perfusion quality are combined with perfusate analysis. The perfusate lactate concentration below a certain threshold is routinely suggested to be of value to predict the outcome. Further cut-offs are suggested by the literature for perfusate pH, transaminases, and biliary parameter. The majority of decisions is based on the combination of such perfusate analyses during NMP (**a**). In contrast, molecules with an origin in mitochondria are used during hypothermic oxygenated liver perfusion with a threshold at FMN 8800 A.U. to avoid graft loss (**b**). Perfusate FMN is routinely combined with NADH concentration, both autofluorescence and measured with a spectrometer during perfusion.

**Table 1 biomedicines-09-00161-t001:** Overview of different technologies available to assess the donor liver quality during donation, preservation, and transplantation.

Detection Method	Material	Parameters in Clinical Use	Time Needed for Assessment	Advantages	Disadvantages
Macroscopic assessment	Entire liver	Size, perfusion quality, steatosis, fibrosis, vessel quality, injuries	Minutes	Routine, rapid, non-invasive, cheap	No information on function, imprecise, assessor dependent
Ultrasound	Entire liver	Size, level of steatosis, lesions	30 min	Easy, assessment of liver parenchyma, rapid, cheap	Assessor dependent, no information on function
Fibroscan	Entire Liver	Level of fibrosis	30 min	Non-invasive, simple, rapid, reproducible, non-operator-dependent	No information on function, additional costs
Histology	Liver tissue	Level of macro and microsteatosis, fibrosis, inflammation	1–2 h	Histological evidence of quality provides criteria to exclude organ transplantation (e.g., Fibrosis)	Invasive, variability in interpretation, biopsy covers only small part of organ, no information on function
Heamodyn-amics during Perfusion	Entire Liver	HA & PV perfusion flow (pressure)	continuous	Real-Time	Not specific for cell type
Blood gas analysis	Perfusate, Effluate, Bile	pO2, pCO2, Lactate, Na, K, pH, Glucose	5–15 min	Non invasive, any type of perfusion, multiple parameters, indirect cholangiocyte assessment, different time points	Timing, different parameters, not specific for a certain cell type
Biochemical analysis	Perfusate, Effluate, Bile	AST, ALT, LDH, HCO3-, ALP	5–15 min	Non-invasive, indirect cholangiocyte assessment, different time points	Not specific for a certain cell type, no functional assessment, no reliable prediction of outcomes after transplantation
Spectroscopy	Perfusate, Effluate	FMN, NADH	5–15 min	Easy, quick, cheap, reliable prediction of graft function, covers entire organ, any type of machine perfusion	No discrimination between different cell types
Metabolomics/proteomics/genomics	Liver tisue, perfusate, effluate, bile	Various molecules from all cellular and sub-cellular compounds	Days/Weeks	Multiple parameters, can be performed in any material (tissue, perfusate, bile)	Requires long time, expensive, not specific for a certain cell type

**Table 2 biomedicines-09-00161-t002:** Viability markers and thresholds defined during machine perfusion of livers for transplantation during the last 5 years (Part 1): the majority of available studies is of retrospective origin and considers donation after circulatory death (DCD) organs. Only a few studies involve a mixed cohort of liver grafts. (* Van Leeuwen series includes the livers from the Matton series; ^#^ Watson series from 2018 includes the series from 2017; graft related relevant complications include PNF, IC or ITBL, AS or leaks, number of EADs was not listed).

Author & Year	Country	Number and Type of Livers	Timepoint’s and Modality	Viability Criteria and Threshold	Prospective Decision Making	Events below the Threshold (within Criteria)
**Hypothermic Machine Perfusion (HOPE, D-HOPE)**						
Schlegel et al., 2020	Switzer-land	50 perfused and transplanted (32 DCD, 18 DBD, HOPE)	Serial measurements (perfusate and tissue) by mass spectometry and spectroscopy, at 15, 30, 60 and 90 min and end of HOPE	Perfusate, tissue and mitochondria: FMN at 30 min (<8800 A.U. or sharp incline), NADH (<10,000 A.U.)	Yes (n = 16/50)	1 ITBL in retrospective, development cohort, none since prospective application of no PNF
Patrono et al., 2020	Italy	50 perfused and transplanted (ECD DBD, D-HOPE)	Every 30 min (3 h perfusion)	Perfusate Lactate, AST, ALT, LDH, glucose, and pH	No	No threshold applied, 3 months follow up
Muller et al., 2019	Switzer-land	100 perfused and transplanted (80 DCD, 20 DBD), 54 assessed, HOPE	Serial perfusate measurements by mass spectometry and spectroscopy, at 15, 30, and 60 min and end of HOPE	Perfusate FMN at 30 min (<8800 A.U.)	No	Threshold established retrospectively
**Normothermic Machine Perfusion (NMP, NMP after COR)**						
Mergental et al., 2020	UK	31 perfused, 22 transplanted (12 DBD, 10 DCD)	Serial measurements every 30 min Assessment at 2.5 and 4 h	Within 4 h of NMP: lactate < 2.5 mmol/L and ≥2 of the following criteria: 1. Evidence of bile production; 2. pH > 7.30; 3. Metabolism of glucose; 4. HA flow > 150 mL/min and PV flow > 500 mL/min; 5. Homogenous perfusion	Yes	2 anastomotic strictures 4 ITBL with retransplantation 6 months follow up
Cardini et al., 2020	Austria	34 perfused, 25 transplanted (21 DBD, 4 DCD)	Serial measurements every hour until 6 h NMP, afterwards every 2 h	1. Rapid decrease and maintenance of lactate levels (first 2 h of NMP); 2. Bile output and biliary pH; 3. Maintenance of physiological perfusate pH without sodium bicarbonate; 4. Exceptionally high OR sharp incline of AST, ALT, LDH	Yes	No PNF 7 anastomotic strictures 3 bile leaks 1 left hemihepatectomy (ascending cholangitis after stricture) 20 months follow up
Zhang et al., 2020	China	4 perfused and transplanted (1 DBD, 3 DCD)	Serial measurements perfusate BGA every 10 min for the first 40 min, and then every 20 min. Bile collected every hour	Within 4 h of NMP: Perfusate lactate ≤ 2.5 mmol/L; 2. Bile production; 3. Perfusate pH ≥ 7.30; 4. Stable HA flow > 150 mL/min and PV flow > 500 mL/min	Yes	1 anastomotic stricture 6 months follow up
Reiling et al., 2020	Australia	10 perfused, transplanted (5 DBD, 5 DCD)	Serial arterial perfusate samples Bile every hour Biopsies at the retrieval, at the end of SCS and at the end of NMP	Within 2 h (to 4 h) of NMP: 1. Lactate clearance to <2 mmol/L 2. Decreasing trend in perfusate glucose concentration by 4 h. 3. Physiological pH without the need for sodium bicarbonate. 4. Stable HA and PV flows. 5. Homogeneous graft perfusion with soft parenchyma consistency 6. Evidence of bile production	Yes	1 Anastomotic leak 1 Anastomotic stricture 6 months follow up
Van Leeuwen et al., 2019	The Netherlands	16 perfused, 11 transplanted (DCD) *	BGA perfusate & Bile: Assessment at 2.5 h (Trial: D-HOPE-COR-NMP)	After 2.5 h of NMP: 1. Lactate clearance to ≤1.7 mmol/L; 2. Perfusate pH 7.35–7.45; 3. Bile production > 10 mL; 4. Biliary pH > 7.45	Yes	1 ITBL 3 Anastomotic strictures Median follow up 12 months
Matton et al., 2019	The Netherlands	(1) 23 perfused to define markers; (2) 6 perfused, 4 transplanted (DCD)	BGA perfusate and Bile: Assessment at 2.5 h (Trial: D-HOPE-COR-NMP)	After 2.5 h of NMP: 1. Lactate clearance to ≤1.7 mmol/L; 2. Perfusate pH 7.35–7.45; 3. Bile production > 10 mL; 4. Biliary pH > 7.48	Yes	No relevant graft-related complications Median follow up 8.3 months
Ceresa et al., 2019	UK	34 perfused, 31 transplanted (23 DBD, 8 DCD)	Perfusate BGA and Bio-chemistry at 15min and 1h of NMP, then every 4 h, and the end of NMP	Within 2.5 h and within 4 h: Lactate clearance and maintenance, glucose metabolism, pH maintenance, bile production, perfusate transaminase levels	Yes	1 Anastomotic leak 1 Anastomotic stricture 12 months follow up
De Vries et al., 2019	The Netherlands	7 perfused, 5 transplanted (DCD)	BGA perfusate & Bile: Assessment at 2.5 h (Trial: D-HOPE-COR-NMP)	After 2.5 h of NMP: 1. Lactate clearance to ≤1.7 mmol/L; 2. Perfusate pH 7.35–7.45; 3. Bile production > 10 mL; 4. Biliary pH > 7.45	Yes	No relevant graft-related complications Median follow up 6.5 months
Watson et al., 2018	UK	47 perfused, 22 transplanted (16 DCD, 6 DBD) ^#^	BGA + Biochemistry: at 10 and 30 min, every 30 min thereafter	1. Peak lactate fall ≥ 4.4 mmol/L/kg/h 2. ALT < 6000 iU/L at 2 h 3. Maximum bile pH > 7.5 4. Bile glucose ≤ 3 mmol/L or 10 mmol less than perfusate glucose 5. Maintain perfusate pH > 7.2 with ≤30 mmol bicarbonate supplementation 6. Falling glucose beyond 2 h OR perfusate glucose < 10 mmol/L with subsequent fall during challenge with 2.5 g glucose	Yes	1 PNF 4 ITBL (3 with retransplantation or awaiting) Median follow up 20 months
Watson et al., 2017	UK	12 perfused and transplanted (9 DCD, 3 DBD)	At 10 and 30 min, every 30 min thereafter	1. Lactate clearance, glucose and transaminase concentrations 2. Maintaining pH without supplemental bicarbonate	No	1 PNF 3 ITBL Median follow up 20 months
Bral et al., 2017	Canada	9 perfused and transplanted 4 DCD, 6 DBD	At perfusion start and every 2 h	pH, Lactate, ALT, AST, bilirubin, perfusion vascular stability, hourly bile production	No	No PNF, one early HCV recurrence with graft loss, 6 months follow up
Mergental et al., 2016	UK	6 perfused 5 transplanted (4 DCD, 1 DBD)	Every 30 min and at 3 h	Within 3 h of NMP: Lactate clearance to <2.5 mmol/L or evidence of bile production combined with two of the following criteria: 1. Perfusate pH > 7.30 2. Hepatic artery flow > 150 mL/min and portal vein flow > 500 mL/min 3. Homogenous perfusion with soft parenchyma consistency	Yes	No relevant graft-related complications Median follow up 7months

## Data Availability

Not applicable.

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
