# Peer review of "Viability Assessment in Liver Transplantation—What Is the Impact of Dynamic Organ Preservation?"

_biomedicines, 2021, doi:10.3390/biomedicines9020161_

Round 1

Reviewer 1 Report

I was very happy to review this manuscript from Panconesi et al. make an extensive review of currently available viability assessment methods during dynamic liver preservation. This is a beautiful manuscript that deserves publication. This is not surprising as some of the authors are pioneers and experts in the field. I will just make a few minor comments which, from my modest point of view, may increase the quality of the manuscript. 

  • From my perspective, the main utility of machine perfusion devices is the rescue of discarded organs, the utility in fatty grafts and in aged ones. I have gone through the manuscript and it is not clear how to handle with these grafts. However, I would really like to know the authors' opinion and the current evidence on the interpretation of these viability tests in different kind of extended criteria grafts. Are the lactate levels the same for a 90% macrosteatotic graft than for a 90 years old one? Shall this interpretation be the same? I would really like to read a specific section about the best ways, interpretation and advices when facing a marginal or discarded graft. 
  • I would suggest authors to review Abbreviations as some of them are not easy to understand and are not routinely used. Most of them appear in figures, so I would suggest to place them in figure legends. For example: FMN, IFOT,...
  • Figure 4. I understand the importance of this figure but so many molecules in the axis is useless. Not sure whether the most important ones could be selected or highlighted,... What do you think?

Again, congratulations to the authors. This is going to become an important manuscript. 

Author Response

Point-by-point reply

Reviewer 1:

I was very happy to review this manuscript from Panconesi et al. make an extensive review of currently available viability assessment methods during dynamic liver preservation. This is a beautiful manuscript that deserves publication. This is not surprising as some of the authors are pioneers and experts in the field. I will just make a few minor comments which, from my modest point of view, may increase the quality of the manuscript. 

  • From my perspective, the main utility of machine perfusion devices is the rescue of discarded organs, the utility in fatty grafts and in aged ones. I have gone through the manuscript and it is not clear how to handle with these grafts. However, I would really like to know the authors' opinion and the current evidence on the interpretation of these viability tests in different kind of extended criteria grafts. Are the lactate levels the same for a 90% macrosteatotic graft than for a 90 years old one? Shall this interpretation be the same? I would really like to read a specific section about the best ways, interpretation and advices when facing a marginal or discarded graft. 

Our Reply:

We thank the reviewer for the kind comments and evaluation of our manuscript.

The reviewer raises a very important point. The majority of viability criteria and thresholds is based on retrospective studies with donation after circulatory death (DCD) livers. And parameters including for example Lactate and Transaminases are generalised to all liver grafts. The available literature on specific markers for steatotic livers or grafts with advanced age is even mor scarce. We have screened the literature again and have specifically highlighted those few retrospective studies, where authors have included different types of livers. We have added a short paragraph on decision making based on viability assessment during NMP (please see 6.7), with a link to the studies listed in Table 2. Additionally, the decision pathway described under point 7 for hypothermic oxygenated perfusion was modified and more details are provided here (please see paragraph before Figure 5: “In clinical practice…”).

  • I would suggest authors to review Abbreviations as some of them are not easy to understand and are not routinely used. Most of them appear in figures, so I would suggest to place them in figure legends. For example: FMN, IFOT,...

Our Reply:

We thank the reviewer for this comment. The abbreviations were included in the specific figure legends, which are marked in the revised manuscript.

  • Figure 4. I understand the importance of this figure but so many molecules in the axis is useless. Not sure whether the most important ones could be selected or highlighted,... What do you think?

Again, congratulations to the authors. This is going to become an important manuscript. 

Our Reply:

We agree with the reviewer, the >2000 molecules appear very “busy” in this figure 4a. We have therefore added a second figure 4b, where the 2000 different molecules are classified into groups. The main message of this figure is, that we are all not aware how many different molecules are circulating in the perfusates during any type of machine perfusion. We have modified the figure legend and explained this in more detail.

Reviewer 2 Report

Dear Authors,

Thank you for allowing me to revise the exciting review article entitled: Viability Assessment in liver transplantation – What is the impact of dynamic organ preservation?

The manuscript is overall interesting and well-written and deals broadly with many of the challenges of graft preservation in the era of machine perfusions.

The problem of assessing the graft's viability is a shared issue for assessing all the machine perfusions since the real task is maintaining the organ functioning. At the same time, many systems cannot assess the functional integrity of the organs.

In this light, two large assessment domains can be distinguished: metabolic preservation and functional preservation. As the authors repetitively stated, metabolic preservation is insufficient to warrant the functional assessment, but functional preservation requires the organ's metabolic and structural integrity.

I kindly ask the authors to review the manuscript spending their efforts to highlight this point, explaining the different challenges and drawbacks to keep the organ well protected both in the normothermic and hypothermic settings.

Minor comments:

Revise the English language as there are some inconsistencies and minor punctuation errors.

Again, compliments for the excellent work

Best Regards

Author Response

Reviewer 2:

Thank you for allowing me to revise the exciting review article entitled: Viability Assessment in liver transplantation – What is the impact of dynamic organ preservation?

The manuscript is overall interesting and well-written and deals broadly with many of the challenges of graft preservation in the era of machine perfusions.

The problem of assessing the graft's viability is a shared issue for assessing all the machine perfusions since the real task is maintaining the organ functioning. At the same time, many systems cannot assess the functional integrity of the organs.

In this light, two large assessment domains can be distinguished: metabolic preservation and functional preservation. As the authors repetitively stated, metabolic preservation is insufficient to warrant the functional assessment, but functional preservation requires the organ's metabolic and structural integrity.

I kindly ask the authors to review the manuscript spending their efforts to highlight this point, explaining the different challenges and drawbacks to keep the organ well protected both in the normothermic and hypothermic settings.

Our Reply:

We thank the reviewer for the kind evaluation of our manuscript and the suggestions.

We agree and consider the point highlighted of great importance. No clear test or parameter has been described yet to discriminate between metabolic and functional organ characteristics because they are tightly connected. We have revised the manuscript and tried to emphasize this fact even more. For both techniques (NP and HMP) we have included a more specific clinical decision pathway (please see point 6.7 and paragraph 7).

Minor comments:

Revise the English language as there are some inconsistencies and minor punctuation errors.

Again, compliments for the excellent work

Our Reply:

We thank the reviewer again for the assessment of our manuscript. All co-authors have revised the final manuscript in this regard and the changes are marked.